# Variations in Orthotropic Elastic Constants of Green Chinese Larch from Pith to Sapwood

**Fenglu Liu** [1,2]**, Houjiang Zhang** [1,2,*]**, Fang Jiang** [1]**, Xiping Wang** [3] **and Cheng Guan** [1,2]

[1]   School of Technology, Beijing Forestry University, Beijing 10083, China; liufenglu3339@bjfu.edu.cn (F.L.); jf0620@bjfu.edu.cn (F.J.); cguan6@bjfu.edu.cn (C.G.)
[2]   Joint International Research Institute of Wood Nondestructive Testing and Evaluation, Beijing Forestry University, Beijing 100083, China
[3]   USDA Forest Products Laboratory, Madison, WI 53726, USA; xiping.wang@usda.gov
[*]   Correspondence: hjzhang6@bjfu.edu.cn; Tel.: +86-010-62336925 (ext. 401)

**Abstract:** Full sets of elastic constants of green Chinese larch (*Larix principis-rupprechtii* Mayr) with 95% moisture content at four different cross-section sampling positions (from pith to sapwood) were determined in this work using three-point bending and compression tests. Variations in the material constants of green Chinese larch from pith to sapwood were investigated and analyzed. The results showed that the sensitivity of each elastic constant to the sampling position was different, and the coefficient of variation ranged from 4.3% to 48.7%. The Poisson's ratios $\nu_{RT}$ measured at four different sampling positions were similar and the differences between them were not significant. The coefficient of variation for Poisson's ratio $\nu_{RT}$ was only 4.3%. The four sampling positions had similar Poisson's ratios $\nu_{TL}$, though the coefficient of variation was 11.7%. The Poisson's ratio $\nu_{LT}$ had the greatest variation in all elastic constants with a 48.7% coefficient of variation. A good linear relationship was observed between the longitudinal modulus of elastic $E_L$, shear modulus of elasticity $G_{RT}$, Poisson's ratio $\nu_{RT}$, and sampling distance. $E_L$, $G_{RT}$, and $\nu_{RT}$ all increased with sampling distance $R$. However, a quadratic relationship existed with the tangential modulus of elasticity $E_T$, radial modulus of elasticity $E_R$, shear modulus of elasticity $G_{LT}$, and shear modulus of elasticity $G_{LR}$. A discrete relationship was found in the other five Poisson's ratios. The results of this study provide the factual changes in the elastic constants of green wood from pith to sapwood for numerical modelling of stress wave propagation in trees or logs.

**Keywords:** orthotropic; elastic constants; green larch; compression; three-point bending

## 1. Introduction

Elastic constants, especially the modulus of elasticity (MOE), which indicate the elastic behavior of wood, are critical parameters for furniture, musical instruments, or wood products, such as plywood, laminated veneer lumber, and cross laminated timber. Numerical simulation is being increasingly used to investigate the propagation of stress wave in standing trees or logs [1–4]. Material elastic constants are required for numerical simulation, especially when defining material properties. Wood, as a complex and anisotropic material, has considerable variations in its mechanical properties from bottom to top, pith to sapwood within a tree. In many studies, wood has been considered an orthotropic material, given its unique and independent material performance in the three principal or orthotropic directions (radial $R$, tangential $T$, and longitudinal $L$) [5–11]. Nine independent elastic constants (reduced from twelve elastic constants according to the symmetry of the stress and strain sensor in orthotropic materials), including three elastic moduli, three shear moduli, and three Poisson's ratios, are required to characterize the elastic behavior of orthotropic materials for mechanical analysis. Wood is also a

hygroscopic material, and its mechanical behavior is therefore impacted by variations in moisture content (MC) [12–16]. Hering et al. determined all the independent elastic properties of European beech wood at different moisture conditions (MC ranged from 8.7% to 18.6%), and results indicated that all elastic parameters, except for Poisson's ratios $v_{TR}$ and $v_{RT}$, show a decrease in stiffness with increasing moisture content [13]. Similarly, Jiang et al. examined the influence of moisture content on the elastic and strength anisotropy of Chinese fir (*Cunninghamia lanceolate* Lamb.) wood and found that, except for Poisson's ratios, all investigated elastic and strength parameters decreased with increasing MC (varied from 10.3% to 16.7%), whereby individual moduli and strength values were affected by the MC to different degrees [14].

Resistance strain gauges and ultrasonic are the two of most commonly used methods determining the elastic constants of wood. As a destructive method, resistance strain gauges are usually used to evaluate the material constants of wood. Sliker first proposed the use of compression and bending tests using strain gauges in wood to determine the elastic constants of materials [6–8]. Many studies had been conducted to verify the feasibility and validity of this method. Li successfully measured the full set of elastic constants for *Fraxinus mandshurica* with 13.4% moisture content via the compression test using strain gauges [9]. Gong estimated the elastic moduli parallel to the grain of *Pinus Massoniana* with 15% MC through strain gauges [10]. Wang et al. determined the full set of material constants for White Birch (*Betula platyphylla* Suk.) with 12% MC using the compression and three-point bending tests [17]. Shao et al. measured the seven elastic constants of *Cunninghamia lanceolata* with 12% MC using electric resistance strain gauges [18]. Aira et al. performed compression tests on dry specimens (around 12% MC) to determine the elastic constants of Scots pine (*Pinus sylvestris* L.) and found the MOE values obtained were greater than the average values for softwoods, and Poisson's ratios obtained parallel to the grain were similar to the values in the literature [19].

Ultrasonic, a rapid and efficient non-destructive method for determining material properties, has drawn increasing attention in wood characteristic measurement. Preziosa et al. first determined the stiffness matrix of wood using the ultrasonic technique [20,21]. Then, Bucur measured the elastic constants of six species (pine, spruce, Douglas-fir, oak, beech, and tulip-tree) in the dry condition, applying ultrasonic to different cubic specimens [22]. Francois proposed the use of a polyhedral specimen with 26 faces for the determination of all the elements in the stiffness matrix from a single specimen to measure the elastic constants of dry wood [23]. Many studies have demonstrated the feasibility of using ultrasonic to measure all the elastic constants from a single specimen of dry wood (MC ranged from 7.5% to 12.3%) from different species, such as *Castanea sativa* Mill., ash (*Fraxinus excelsior* L), beech, *Eucalyptus saligna*, *Apuleia leiocarpa*, and *Goupia glabra* [24–27].

Although the full material constants of wood can be measured by both resistance strain gauges and ultrasonic, the full sets of elastic constants for green wood have rarely been reported in the literature. As wood is often used in a dry state, the elastic constants reported in most published papers for wood are in these conditions [28–31]. This would lead to a poor representation when the elastic constants of dry wood are used for modelling standing trees or green logs. Only Davies et al. investigated and obtained the elastic constants of green *Pinus radiata* wood using compression and tension tests [32]. Davies et al. stated that the mathematical modelling would be more realistic with the material constants of green wood rather than those of dry wood. The variation in elastic constants in cross-sections of green wood from pith to sapwood have not been reported in the literature. Davies et al. only measured the material constants of the corewood and outerwood of *Pinus radiata* instead of elastic constants across whole transverse sections of wood. Although Xavier et al. reported that the two stiffness values ($Q_{22}$ and $Q_{66}$) of *P. pinaster* varied across three or four different radial positions using the unnotched Iosipescu test, the moisture content of the specimens was 10.4% and only two stiffness values were investigated [33]. Therefore, to the best of our knowledge, no full sets of elastic constants varying from pith to sapwood have been published for green wood. The variations in the mechanical properties of green wood from pith to sapwood have not yet been studied.

The main purpose of this research was to determine the elastic constants of green Chinese larch from pith to sapwood using compression and three-point bending tests, as well as to investigate and analyze the variations in the mechanical characteristics of green Chinese larch from pith to sapwood. We aimed to obtain basic knowledge about the mechanical properties of wood from pith to sapwood, and to describe standing trees or logs as an orthotropic material in numerical modelling. Use of wood could be optimized in various applications, such as papermaking, furniture, musical instruments, or wood products, according to the measured material performance in different parts of the wood. Standing trees or logs could be modelled more realistically in numerical simulation with the acquired data. The results of this study provide factual elastic constants of green wood from pith to sapwood for numerical modelling of stress wave propagation in trees or logs.

## 2. Materials and Methods

### 2.1. Materials

Two Chinese larches (*Larix principis-rupprechtii* Mayr), a common plantation species in Northern China, were harvested from Maojingba National Plantation Farm, located in Longhua County, Chengde City, Hebei Province, China (118°06′05″ E, 41°28′46″ N at approximately 750 m elevation). The trees (coded A and B) aged 40 years were felled and branches were subsequently removed. The diameter at breast height (DBH) values of tree A and tree B were 32 cm and 36 cm, respectively. Then, two 60-cm-long logs were cut from each selected tree at a height of 0.5 m and 1.25 m above the ground. A 15-cm-thick disc for density and moisture content measurements was cut from each tree at a height of 1.1 m above the ground. A total of four 60-cm-long logs and two 15-cm-thick discs were obtained and immediately sealed in plastic wrap. After, these logs and discs were directly transported to the mechanics laboratory in Beijing Forestry University and kept in a condition room at 15 °C and 95% relative humidity.

### 2.2. Static Testing Method

#### 2.2.1. Specimen Sampling

A schematic of the sawing pattern used to obtain green larch specimens for static testing is presented in Figure 1. Four different sampling positions from pith to sapwood (numbered 1, 2, 3, and 4 in Figure 1 referred to sampling position, defined as P1, P2, P3, and P4 hereinafter, respectively) were chosen to determine the elastic constants at different positions from pith to sapwood in the cross-sections of standing trees and investigate the distribution of elastic constants in the cross-sections of standing trees. As shown in Figure 1, sampling positions P1, P2, and P4 were located at the pith, heartwood, and sapwood, respectively, whereas P3 was located between the heartwood and sapwood. The initial transverse dimension of specimens for static testing in each sampling location was 53 × 53 mm.

Parameter *R* was used to define the distance between the center of pith and the center of sampling position. Thus, the designed distance *R* for sampling position P1, P2, P3 and P4 was 13 mm, 56 mm, 76 mm and 132 mm, respectively. It should be noticed that the sampling distance *R* for P1, P2, P3 and P4 shown in Figure 1 was designed for a log with a DBH ranging from 300 mm to 360 mm. The sampling distance would be different as the DBH of log over 360 mm and need to be changed.

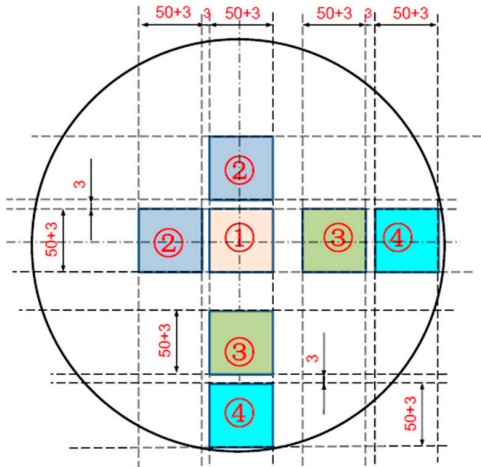

**Figure 1.** Schematic of sawing pattern for testing specimens. Dashed lines represent the sawing line and the units are millimeters. The extra 3 mm width was the machining allowance. The number 1, 2, 3, and 4 in the yellow, gray, green, and blue area, respectively represent four different sampling positions.

One P1 lumber (53 × 53 × 600 mm), two P2 lumbers (53 × 53 × 600 mm), two P3 lumbers (53 × 53 × 600 mm), and two P4 lumbers (53 × 53 × 600 mm) were obtained from each log according to sampling pattern shown in Figure 1. A total of 4 P1- lumbers, 8 P2- lumbers, 8 P3- lumbers, and 8 P4- lumbers were acquired from four 600-mm-long logs. Then, these were used to prepare the specimens for static testing, including the compression test and three-point bending test, performed according to American Society for Testing Materials (ASTM) D5536-94 [34]. Specimens for static testing were instantly sealed with plastic wrap and stored in the condition room (15 °C and 95% relative humidity) before mechanical testing. The dimensions of the specimens used for static testing are provided in Table 1. Each sampling location (referred to as P1, P2, P3, and P4) all used the same size specimens for static testing. Therefore, the elastic constants of these four different positions (from pith to sapwood) in the cross-section of standing trees could be determined.

**Table 1.** Dimensions of specimens for static testing.

| Static Test | Orientation | | Size (mm) (Length × Width × Thickness) | Number of Specimens for One Sampling Position |
| --- | --- | --- | --- | --- |
| Compression test | Parallel to grain | Longitudinal (L) | 25 × 25 × 100 | 8 |
| | Perpendicular to grain | Radial (R) Tangential (T) With a 45° to tangential | 50 × 50 × 150 | 8 × 3 |
| Three-point bending test | Parallel to grain | Longitudinal (L) | 25 × 25 × 150 | 8 |
| | | | 25 × 25 × 200 | 8 |
| | | | 25 × 25 × 250 | 8 |
| | | | 25 × 25 × 300 | 8 |
| | | | 25 × 25 × 350 | 8 |

### 2.2.2. Compression Test

The four types of test specimens used for the compression test, with wood grain oriented relative to the orthotropic directions and the distribution of strain gauges in specimen, are displayed in Figure 2. Eight clear test specimens of the required shape and orientation were machined from the cut lumber. A total of 128 test specimens (8 replicates × 4 orientations × 4 sampling positions) were used for compression testing. Eight 25 × 25 × 100 mm specimens parallel to the grain (Figure 2a) were used to measure the elastic constants of $E_L$, $\nu_{LR}$, and $\nu_{LT}$. Eight 50 × 50 × 150 mm specimens perpendicular to the grain radially (Figure 2b) were used to test $E_R$, $\nu_{RL}$, and $\nu_{RT}$. Eight 50 × 50 × 150 mm specimens perpendicular to the grain tangentially (Figure 2c) were used to evaluate $E_T$, $\nu_{TR}$, and $\nu_{TL}$.

Eight $50 \times 50 \times 150$ mm specimens inclined at a 45° to the grain (Figure 2d) were used to obtain the shear modulus of elasticity $G_{RT}$.

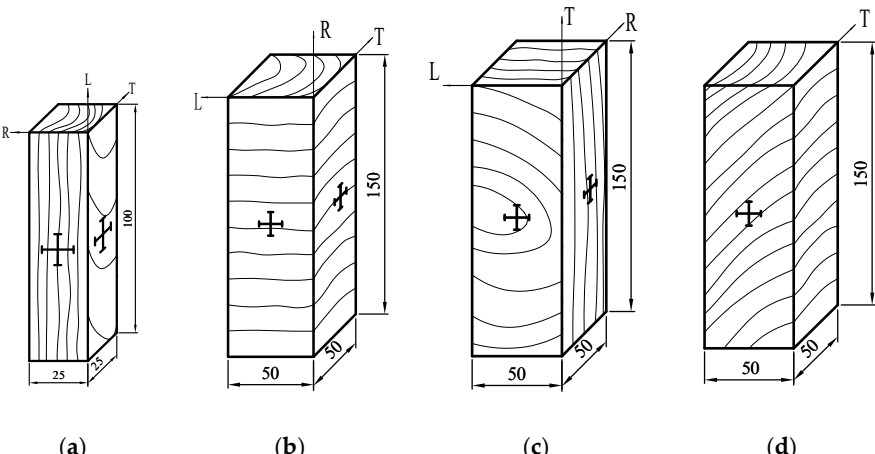

(a)        (b)        (c)        (d)

**Figure 2.** Types of specimens and strain gauges placement for compression test: (**a**) parallel to the grain, (**b**) perpendicular to the grain radially, (**c**) perpendicular to the grain tangentially, and (**d**) inclined at 45°.

Resistance strain gauges were directly bonded to the surfaces of the specimen prior to compression testing. Figure 2a–c show that four strain gauges were placed on each specimen. Two resistance strain gauges were glued perpendicularly to a surface of testing sample, and the other two resistance strain gauges were perpendicularly bonded to the adjacent surface. As shown in Figure 2d, only two strain gauges perpendicularly glued to a surface of specimen were used for compression testing. Strain gauges should be placed close to each other so that measurements record the strain state of the same point with no variability in their elastic properties due to the heterogeneity of wood. The adhesive used has a lower longitudinal stiffness than wood to avoid restricting its free deformation when receiving the external load. The adhesive must simultaneously have a high shear stiffness so that the deformation of strain gauge is not damped by the thickness of the adhesive [19]. RGM-4050-100 (made by Reger Instrument Corporation Limited, Shenzhen City, China), a microcomputer-controlled electronic universal testing machine, was used to conduct the compression test according to ASTM D143-09 [35]. Specimens were loaded at a rate of 0.2 mm/min for compression testing. Modulus of elasticity ($E_L$, $E_R$, and $E_T$), Poisson's ratios ($v_{LR}$, $v_{LT}$, $v_{RL}$, $v_{RT}$, $v_{TR}$, and $v_{TL}$) and shear modulus of elasticity ($G_{RT}$) were calculated from Equations (1)–(3), respectively.

$$E = \frac{P_n - P_0}{A_0(\varepsilon_n - \varepsilon_0)} \tag{1}$$

where $E$ is the modulus of elasticity (MPa), $P_n$ is the final load (N), $P_0$ is the initial load (N), $A_0$ is the cross-section area of specimen (mm$^2$), $\varepsilon_n$ is the final strain, and $\varepsilon_0$ is the initial strain.

$$v = -\frac{\Delta \varepsilon'}{\Delta \varepsilon} \tag{2}$$

where $v$ is the Poisson's ratio of specimen, $\Delta \varepsilon'$ is the lateral strain increase, and $\Delta \varepsilon$ is the axial strain increase.

$$G_{RT} = \frac{\Delta P_{45°}}{2A_0\left(\Delta \varepsilon_x - \Delta \varepsilon_y\right)} \tag{3}$$

where $G_{RT}$ is the shear modulus of elasticity in the RT plane (MPa), $A_0$ is the cross-section area of the specimen (mm$^2$), $\Delta P_{45°}$ is the load increase of the elastic deformation phase on the load-strain

curve (N), $\Delta\varepsilon_x$ is the strain increase along the axis of the specimen, and $\Delta\varepsilon_y$ is the strain increase perpendicular to the axis of specimen.

Since the moisture content has a significant effect on the mechanical properties of wood, the specimens were taken from the condition room and tested in sequence to ensure less variation in the moisture content of specimens. Measurements for each specimen were performed as soon as possible to reduce the impact of the moisture content on the testing results.

### 2.2.3. Three-Point Bending Test

In the light of ASTM D5536-94, specimens with five different ratios of span to depth were prepared to conduct three-point bending test. We obtained a total of 160 test specimens in total (8 replicates × 5 spans × 4 sampling positions) for the bending tests. The specific size of the specimens, especially the length, are shown in Table 1. The span for each length of specimen was 132 mm, 176 mm, 220 mm, 264 mm, and 308 mm, respectively. Three-point bending tests were conducted using an RGM-4050-100 universal testing machine on the basis of ASTM D143-09. The loading speed of specimens for three-point bending test was 5 mm/min. As above, the specimens for the bending test were successively measured and tested as quickly as possible to reduce the variation in the moisture content of the measured samples. Bending moduli of elasticity (*MOE*) were calculated using Equation (4), and then shear modulus of elasticity (*G*) can be obtained using Equation (5).

$$MOE = \frac{\Delta P l^3}{4\Delta f b h^3} \tag{4}$$

where *MOE* is the bending modulus of elasticity (MPa), $\Delta P$ is loading increase (N), $\Delta f$ is the deflection increase (mm), *b* is the width of the specimen (mm), *h* is the thickness of the specimen (mm), and *l* is the span of the specimen (mm).

$$G = \frac{1.2\Delta(h/l)^2}{\Delta(1/MOE)} \tag{5}$$

where $\Delta(h/l)^2$ is the increase in the square of ratio between the thickness and span and $\Delta(1/MOE)$ is the increase in the reciprocal of the bending modulus of elasticity (mm$^2$/N). The shear modulus of elasticity $G_{LR}$ was obtained by measuring specimen loaded from radial direction, and the shear modulus of elasticity $G_{LT}$ was obtained by measuring the specimen loaded from tangential direction. Both of them can be calculated using Equations (4) and (5).

## 3. Results and Discussion

### 3.1. Twelve Elastic Constants of Green Chinese Larch at Different Sampling Positions

The moisture content (MC) of green Chinese larch was measured in the laboratory using the kiln-dry method, and the average MC of the sample trees was 95%. The average green density of Chinese larch was 625 kg/m$^3$. Twelve elastic constants of green Chinese larch were calculated using Equations (1)–(4) using the experimental data obtained from three-point bending and compression tests. The elastic constants of the four different sampling positions (P1, P2, P3, and P4) are shown in Table 2. The average values of the elastic moduli in the longitudinal, radial, and tangential directions were 7,629 MPa, 773 MPa, and 362 MPa, respectively. Davies et al. reported that the longitudinal, radial, and tangential elastic moduli of the outerwood in green *Pinus radiata* were 4,360 MPa, 490 MPa, and 250 MPa, respectively. The values for the corewood of green *Pinus radiata* were 3,500 MPa, 260 MPa, and 240 MPa, respectively. The average values of the three elastic moduli obtained from this work were higher than those derived from Davies's research due to the differences in the tested species and tree ages. The longitudinal modulus of elasticity ($E_L$) increased from 5016 MPa to 10,137 MPa as the sampling distance (*R*) varied from 13 mm (P1) to 132 mm (P4), respectively. This means that the longitudinal mechanical properties of green Chinese larch increased from pith to sapwood. However, the radial

modulus of elasticity ($E_R$) initially increased from 628 MPa to 1,154 MPa as the sampling distance changed from 13 mm (P1) to 76 mm (P3) and then decreased to 342 MPa as sampling distance increased to 132 mm (P4). This may indicate a lower radial modulus of elasticity in sapwood. No significant relationship was found between tangential modulus of elasticity ($E_T$) and sampling distance ($R$).

The average values of shear moduli in the LR, LT, and RT plane were 428 MPa, 393 MPa, and 450 MPa, respectively. Davies et al. presented the lower values for these three shear moduli, 60 MPa, 110 MPa, 50 MPa in LR, LT, and RT plane for outerwood and 40 MPa, 110 MPa, 20 MPa in LR, LT, and RT plane for corewood, respectively. The average values of $\nu_{LT}$, $\nu_{LR}$, $\nu_{TL}$, $\nu_{TR}$, $\nu_{RL}$, and $\nu_{RT}$ Poisson's ratios were 0.30, 0.22, 0.04, 0.60, 0.05, and 0.77, respectively. The corresponding values of these six Poisson's ratios in Davies's paper were 0.60, 0.29, 0.03, 0.33, 0.03, and 0.64 for outerwood and 0.16, 0.46, 0.01, 0.33, 0.05, and 0.54 for corewood, respectively. However, no significant relationship with sampling distance was found for either shear modulus of elasticity or Poisson's ratio.

**Table 2.** Twelve elastic constants at different sampling locations in cross-sections of green Chinese larch.

| Sampling Position | $E_L$ (MPa) | $\nu_{LT}$ | $\nu_{LR}$ | $E_T$ (MPa) | $\nu_{TL}$ | $\nu_{TR}$ | $E_R$ (MPa) | $\nu_{RL}$ | $\nu_{RT}$ | $G_{RT}$ (MPa) | $G_{LR}$ (MPa) | $G_{LT}$ (MPa) |
|---|---|---|---|---|---|---|---|---|---|---|---|---|
| 1 | 5016 | 0.15 | 0.13 | 339 | 0.04 | 0.55 | 628 | 0.04 | 0.74 | 326 | 538 | 403 |
| 2 | 5996 | 0.47 | 0.34 | 423 | 0.05 | 0.66 | 967 | 0.03 | 0.75 | 535 | 339 | 352 |
| 3 | 9365 | 0.21 | 0.15 | 402 | 0.04 | 0.53 | 1154 | 0.06 | 0.79 | 383 | 404 | 371 |
| 4 | 10137 | 0.36 | 0.26 | 282 | 0.04 | 0.66 | 342 | 0.05 | 0.81 | 556 | 430 | 446 |
| Mean | 7629 | 0.30 | 0.22 | 362 | 0.04 | 0.60 | 773 | 0.05 | 0.77 | 450 | 428 | 393 |
| SD [1] | 2503 | 0.15 | 0.10 | 64 | 0.01 | 0.07 | 360 | 0.01 | 0.03 | 113 | 83 | 41 |
| SE [2] | 1252 | 0.07 | 0.05 | 32 | 0.01 | 0.04 | 180 | 0.01 | 0.02 | 57 | 41 | 21 |
| COV [3] (%) | 32.8 | 48.7 | 44.7 | 17.7 | 11.7 | 11.6 | 46.6 | 28.7 | 4.3 | 25.1 | 19.4 | 10.5 |

[1] SD, standard deviation; [2] SE, standard error; [3] COV, coefficient of variation.

Table 2 shows that the longitudinal modulus of elasticity ($E_L$) was higher than the radial modulus of elasticity ($E_R$), and the radial modulus of elasticity ($E_R$) was greater than the tangential modulus of elasticity ($E_T$), i.e., $E_L > E_R > E_T$, for all four sampling positions. Similarly, Poisson's ratio $\nu_{RT}$ was higher than Poisson's ratio $\nu_{LT}$, and Poisson's ratio $\nu_{LT}$ was greater than Poisson's ratio $\nu_{LR}$, i.e., $\nu_{RT} > \nu_{LT} > \nu_{LR}$ at these four sampling locations. These results align with the findings in dry wood [17,18]. Wood is a highly anisotropic material. Thus, different values would be obtained for the same elastic constant due to different sampling positions. Table 2 also shows that the sensitivity of each elastic constant to the sampling position was different, and the corresponding coefficient of variation ranged from 4.3% to 48.7%. Table 3 provides the results of the analysis of variance (ANOVA) for each elastic constant at different sampling positions. Table 3 shows that sampling position had a significant impact both on $E_L$, $G_{RT}$, $\nu_{LT}$, and $\nu_{LR}$ ($p < 0.05$), whereas no significant effect was found in the other elastic constants ($p > 0.05$). For three MOE, only $E_L$ showed significant differences ($p < 0.05$) in sampling positions with a 32.8% coefficient of variation. Poisson's ratios $\nu_{RT}$ measured at four different sampling positions were similar and the coefficient of variation for Poisson's ratio $\nu_{RT}$ was only 4.3%, which is in agreement with the insignificant differences between them ($p > 0.05$). The four sampling positions had similar Poisson's ratios $\nu_{TL}$ and showed an insignificant difference ($p > 0.05$), though the coefficient of variation was 11.7%. Poisson's ratio $\nu_{LT}$ had the greatest variation in all elastic constants with a 48.7% coefficient of variation and showed a significant difference in sampling position ($p < 0.05$). For shear moduli, only $G_{RT}$ showed significant differences ($p < 0.05$) in sampling positions with a 25.1% coefficient of variation.

**Table 3.** Analysis of variance of elastic constants at different sampling positions.

| | Source | Sum of Squares | Degrees of Freedom | Mean Square | F-Value | p-Value |
|---|---|---|---|---|---|---|
| $E_L$ | Between groups [1] | $9.92 \times 10^7$ | 3 | $3.31 \times 10^7$ | 8.31 | 0.001 |
| | Within groups | $9.55 \times 10^7$ | 24 | $3.98 \times 10^6$ | | |
| | Total | $1.95 \times 10^8$ | 27 | | | |
| $E_R$ | Between groups | $1.99 \times 10^6$ | 3 | $6.65 \times 10^5$ | 1.62 | 0.21 |
| | Within groups | $9.84 \times 10^6$ | 24 | $4.09 \times 10^5$ | | |
| | Total | $1.18 \times 10^7$ | 27 | | | |
| $E_T$ | Between groups | $1.55 \times 10^6$ | 3 | $5.17 \times 10^5$ | 1.07 | 0.38 |
| | Within groups | $1.16 \times 10^7$ | 24 | $4.82 \times 10^5$ | | |
| | Total | $1.31 \times 10^7$ | 27 | | | |
| $G_{LR}$ | Between groups | $8.27 \times 10^4$ | 3 | $2.76 \times 10^4$ | 1.29 | 0.322 |
| | Within groups | $2.56 \times 10^5$ | 12 | $2.13 \times 10^4$ | | |
| | Total | $3.38 \times 10^6$ | 15 | | | |
| $G_{LT}$ | Between groups | $2.04 \times 10^4$ | 3 | $6.79 \times 10^4$ | 0.49 | 0.694 |
| | Within groups | $1.65 \times 10^5$ | 12 | $1.38 \times 10^4$ | | |
| | Total | $1.86 \times 10^5$ | 15 | | | |
| $G_{RT}$ | Between groups | $2.23 \times 10^5$ | 3 | $7.43 \times 10^4$ | 25.57 | $3.25 \times 10^{-4}$ |
| | Within groups | $5.52 \times 10^4$ | 19 | $2.91 \times 10^3$ | | |
| | Total | $2.78 \times 10^5$ | 22 | | | |
| $v_{LT}$ | Between groups | 0.439 | 3 | 0.146 | 12.14 | $1.54 \times 10^{-3}$ |
| | Within groups | 0.290 | 24 | 0.012 | | |
| | Total | 0.729 | 27 | | | |
| $v_{LR}$ | Between groups | 0.221 | 3 | 0.074 | 12.82 | $1.12 \times 10^{-3}$ |
| | Within groups | 0.138 | 24 | 0.006 | | |
| | Total | 0.358 | 27 | | | |
| $v_{TL}$ | Between groups | 0.006 | 3 | 0.002 | 0.644 | 0.594 |
| | Within groups | 0.072 | 24 | 0.003 | | |
| | Total | 0.078 | 27 | | | |
| $v_{TR}$ | Between groups | 0.105 | 3 | 0.035 | 1.505 | 0.239 |
| | Within groups | 0.558 | 24 | 0.023 | | |
| | Total | 0.663 | 27 | | | |
| $v_{RL}$ | Between groups | 0.005 | 3 | 0.002 | 0.974 | 0.421 |
| | Within groups | 0.045 | 24 | 0.002 | | |
| | Total | 0.050 | 27 | | | |
| $v_{RT}$ | Between groups | 0.024 | 3 | 0.008 | 0.187 | 0.904 |
| | Within groups | 1.040 | 24 | 0.043 | | |
| | Total | 1.065 | 27 | | | |

[1] Groups: there are four groups, i.e., four different sampling positions P1, P2, P3, and P4.

### 3.2. Vadility of Measured Data

Although the elastic constants of green Chinese larch at the four different sampling positions were obtained through experiments and data processing, the validity of testing data needed further verification. According to the mechanics of composite materials, the elastic constants of orthotropic materials should satisfy the limitations of Maxwell's theorem, as shown in Equations (6) and (7). As mentioned, in many studies, wood is considered an orthotropic material in the three main orthotropic directions. Therefore, the modulus of elasticity and the Poisson's ratio of green larch measured in this research should be satisfied the limitations of Maxwell's theorem.

$$\frac{v_{ij}}{E_i} = \frac{v_{ji}}{E_j} (i, j = \mathrm{L, R, T}) \tag{6}$$

$$|v_{ij}| < \left| \frac{E_i}{E_j} \right|^{\frac{1}{2}} \tag{7}$$

The modulus of elasticity and the Poisson's ratio obtained from P1 sampling position were taken into Equations (6) and (7), and the results showed that these elastic constants from P1 satisfy the limitation of Maxwell's theorem. Similar results were found in the P2, P3, and P4 sampling locations. Table 4 provides the results of Equation (7) at the P1, P2, P3, and P4 sampling locations. These results indicate that acquired data and calculated elastic constants from compression test were both valid and accurate.

**Table 4.** Results of the limitations of Maxwell's theorem at the four different sampling locations.

| Sampling Position | $\lvert v_{LT} \rvert$ | $\lvert \frac{E_L}{E_T} \rvert^{\frac{1}{2}}$ | $\lvert v_{LR} \rvert$ | $\lvert \frac{E_L}{E_R} \rvert^{\frac{1}{2}}$ | $\lvert v_{TL} \rvert$ | $\lvert \frac{E_T}{E_L} \rvert^{\frac{1}{2}}$ | $\lvert v_{TR} \rvert$ | $\lvert \frac{E_T}{E_R} \rvert^{\frac{1}{2}}$ | $\lvert v_{RL} \rvert$ | $\lvert \frac{E_R}{E_L} \rvert^{\frac{1}{2}}$ | $\lvert v_{RT} \rvert$ | $\lvert \frac{E_R}{E_T} \rvert^{\frac{1}{2}}$ |
|---|---|---|---|---|---|---|---|---|---|---|---|---|
| 1 | 0.15 | 3.85 | 0.13 | 2.83 | 0.04 | 0.26 | 0.55 | 0.73 | 0.04 | 0.35 | 0.74 | 1.36 |
| 2 | 0.47 | 3.76 | 0.34 | 2.49 | 0.05 | 0.27 | 0.66 | 0.66 | 0.03 | 0.40 | 0.75 | 1.51 |
| 3 | 0.21 | 4.83 | 0.15 | 2.85 | 0.04 | 0.21 | 0.53 | 0.59 | 0.06 | 0.35 | 0.79 | 1.69 |
| 4 | 0.36 | 6.00 | 0.26 | 5.44 | 0.04 | 0.17 | 0.66 | 0.91 | 0.05 | 0.18 | 0.81 | 1.10 |

However, Maxwell's theorem was only used to verify the validity of the modulus of elasticity and Poisson's ratio and could not be used to confirm the validity of the shear modulus of elasticity. Correlation analysis, consequently, was conducted on experimental data from the three-point bending tests to verify the validity of calculated shear modulus of elasticity. The relationship between the square of ratio between thickness and span $(h/l)^2$ and the reciprocal of bending modulus of elasticity $(1/MOE)$ was analyzed both for radial-loaded and tangential-loaded bending tests. The results of correlation analysis for the P1, P2, P3, and P4 sampling positions are presented in Figures 3–6, respectively.

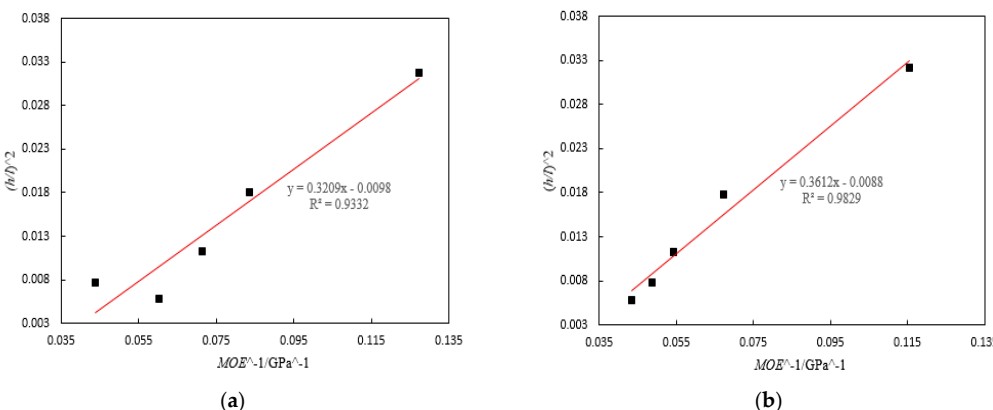

(a)                                        (b)

**Figure 3.** Relationship between the reciprocal of bending elastic modulus and the square of the ratio between depth and length (Sampling position 1) for: (**a**) tangential loading and (**b**) radial loading.

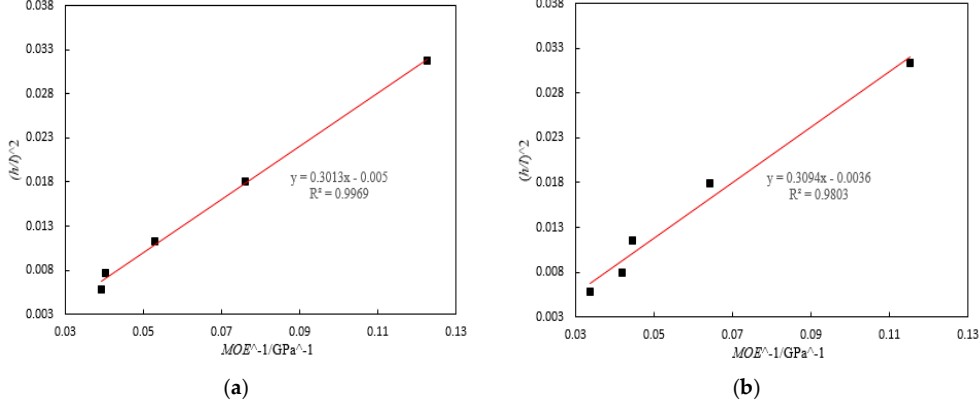

(a)                                        (b)

**Figure 4.** Relationship between the reciprocal of bending elastic modulus and the square of the ratio between depth and length (Sampling position 2) for: (**a**) tangential loading and (**b**) radial loading.

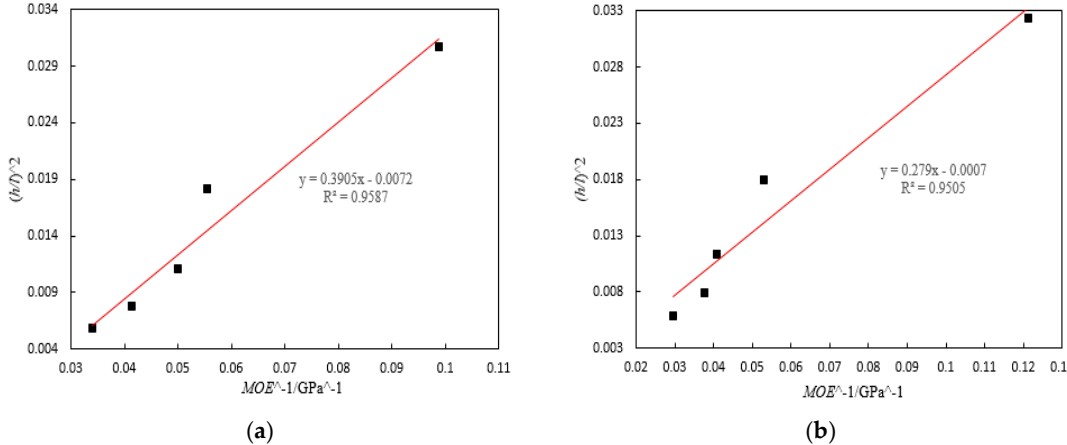

**Figure 5.** Relationship between the reciprocal of bending elastic modulus and the square of the ratio between depth and length (Sampling position 3) for: (**a**) tangential loading and (**b**) radial loading.

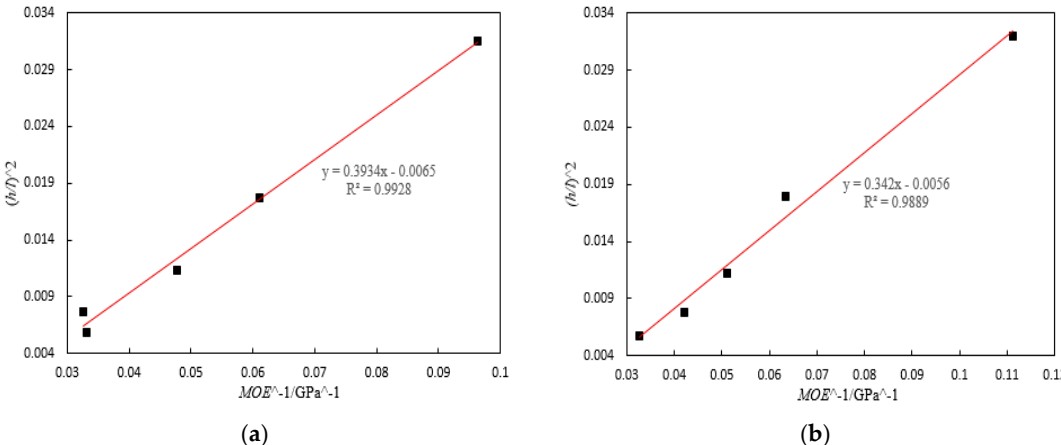

**Figure 6.** Relationship between the reciprocal of bending elastic modulus and the square of the ratio between depth and length (Sampling position 4) for: (**a**) tangential loading and (**b**) radial loading.

Figures 3–6 show that a linear relationship between the square of ratio in span and depth and the reciprocal of bending modulus of elasticity was found in the four sampling positions both for tangential-loaded and radial-loaded bending tests. The correlation coefficients between the square of the ratio in span and depth and the reciprocal of bending modulus of elasticity for the P1, P2, P3, and P4 sampling positions were all over 0.9 in the tangential-loaded and radial-loaded bending tests. These results indicate that the three-point bending test data and calculated shear modulus of elasticity were both effective and reasonable.

*3.3. Variation in Elastic Constants of Wood Cross-Sections*

3.3.1. Modulus of Elasticity

Wood is a highly anisotropic material with different mechanical properties throughout its interior. Wood properties change from pith to bark within a tree and differ between trees. Therefore, the mechanical properties, especially the elasticity constants, of wood vary along the cross-section. To investigate the difference and variation in the elastic constants along the cross-section of wood, the relationships between the modulus of elasticity, shear modulus of elasticity, Poisson's ratios, and sampling distance $R$ were analyzed by regression analysis to obtain the variation patterns of the elastic constants along the cross-section of green Chinese larch using the experimental data derived from compression and three-point bending tests.

The results of regression analysis for modulus of elasticity and sampling position are provided in Figure 7 and Table 5. Relationships between sampling distance and the modulus of elasticity for the three principal axes of wood are illustrated in Figure 7. Table 5 displays the corresponding fitting equations and correlation coefficients came from regression analysis.

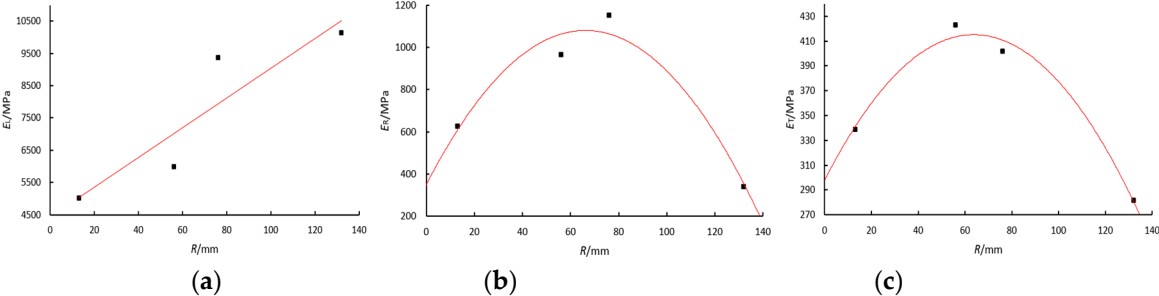

**Figure 7.** Relationships between the distance from pith and elastic moduli in the (**a**) longitudinal, (**b**) radial, and (**c**) tangential directions.

**Table 5.** Mathematical model of elastic moduli in three principle directions and distance from pith.

| Modulus of Elasticity | Fitted Equation | Coefficient of Determination ($R^2$) |
|---|---|---|
| $E_L$ | $E_L = 4436.36 + 46.12R$ | 0.91 |
| $E_R$ | $E_R = 343.25 + 22.27R - 0.17R^2$ | 0.95 |
| $E_T$ | $E_T = 297.55 + 3.68R - 0.03R^2$ | 0.98 |

Figure 7 shows the variation patterns of the three principal moduli of elasticity along the cross-section of the wood. Figure 7a shows that the longitudinal modulus of elasticity ($E_L$) of green Chinese larch linearly increased with sampling distance. However, a quadratic relationship was observed between the radial modulus of elasticity ($E_R$) and the sampling distance, as well as for the tangential modulus of elasticity ($E_T$) and the sampling distance. $E_R$ and $E_T$ both first increased with sampling distance, and then decreased with sampling distances over 70 mm, as shown in Figure 7b,c. $E_R$ and $E_T$ near the bark were significantly lower than in other sampling positions, and even lower than the measured values near the pith. Table 5 shows the linear relationship between the longitudinal modulus of elasticity and sampling distance ($R^2 = 0.91$). Even though a quadratic relationship was found in the tangential and radial moduli of elasticity, both coefficients of determination were higher than 0.95. Little research has been conducted to investigate the variation in elastic constants of wood from pith to sapwood. Only Xavier et al. studied the variation in two stiffness values ($Q_{22}$ and $Q_{66}$) of dry Maritime pine across the radial position using the unnotched Iosipescu test. They found the transverse stiffness ($Q_{22} = E_R/(1 - v_{LR}.v_{RL})$) of dry Maritime pine decreased between the radial position $r_1$ (thirteenth ring, 29% of the radius) and $r_2$ (nineteenth ring, 46% of the radius), and a progressive increase was observed up to $r_4$ (forty-third ring, 81% of the radius) [33]. This means that the transverse stiffness decreased from the center to about the middle radius of stem and increased afterward to the outermost positions. The variation pattern of elastic moduli $E_R$ of green Chinese larch measured in this work was different from that of the transverse stiffness $Q_{22}$. This may be because the transverse stiffness was not only affected by the radial elastic moduli but also by Poisson's ratios $v_{LR}$ and $v_{RL}$. Different moisture contents and tree species may also produce these differences. More data from the same or different species should be acquired to investigate the variation in these three elastic constants in dry or green wood.

### 3.3.2. Shear Modulus of Elasticity

The results of regression analysis for the shear modulus of elasticity and sampling position are provided in Figure 8 and Table 6. The relationships between sampling distance and the shear modulus

of elasticity are illustrated in Figure 8. Table 6 provides the corresponding fitting equations and correlation coefficients.

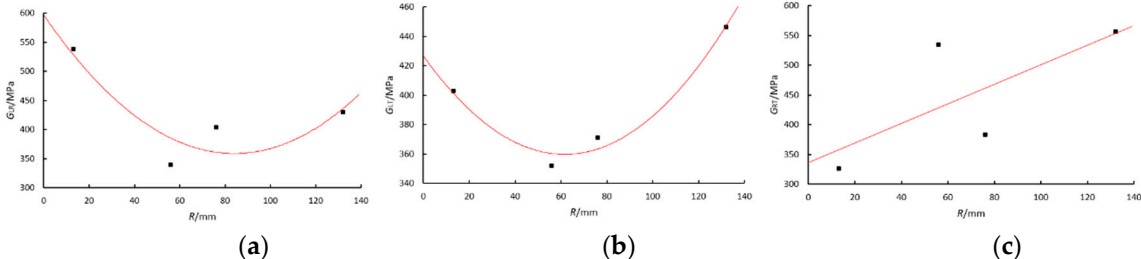

**Figure 8.** Relationships between distance from pith and shear moduli in: (**a**) Longitudinal-radial (LR) plane, (**b**) Longitudinal-tangential (LT) plane, and (**c**) Radial-tangential (RT) plane.

**Table 6.** Mathematical model of shear moduli and distance from pith.

| Shear Modulus of Elasticity | Fitted Equation | Coefficient of Determination ($R^2$) |
|---|---|---|
| $G_{LR}$ | $G_{LR} = 597.134 + 64.293R + 0.034R^2$ | 0.80 |
| $G_{LT}$ | $G_{LT} = 426.009 - 2.155R + 0.018R^2$ | 0.98 |
| $G_{RT}$ | $G_{RT} = 335.912 + 1.645R$ | 0.72 |

Figure 8 shows the variation patterns of the three shear moduli of elasticity ($G_{LR}$, $G_{LT}$, and $G_{RT}$) along the cross-section of wood. Figure 8a,b demonstrate a quadratic relationship between shear modulus of elasticity $G_{LR}$ and sampling distance, as well as for shear modulus of elasticity $G_{LT}$ and sampling distance. Both $G_{LR}$ and $G_{LT}$ first decreased with increasing sampling distance, and then increased at a sampling distance over 70 mm. $G_{LR}$ and $G_{LT}$ near the bark increased compared to the minimum values. However, the shear modulus of elasticity $G_{RT}$ of green Chinese larch linearly increased with sampling distance. Table 6 shows the coefficient of determination for the shear modulus of elasticity $G_{RT}$ and the sampling distance was 0.72, indicating a robust linear relationship between them. Even though a quadratic relationship was found in the shear modulus of elasticity $G_{LR}$ and $G_{LT}$, their coefficients of determination were 0.80 and 0.98, respectively. The possible interpretation for the relatively lower coefficient of determination ($R^2$) for $G_{RT}$ could be attributed to the large variability in wood performance especially in the cross-sections. Xavier et al. reported the shear stiffness ($Q_{66} = G_{LR}$) of dry Maritime pine decreased from the center (radial position $r_1$, 29% of the radius) to around the middle radius of stem (radial position $r_2$, 46% of the radius), and progressively increased afterward to the outermost positions ($r_4$, 81% of the radius) [33]. Despite the different moisture content and tree species, the variation in shear moduli $G_{LR}$ along the whole cross-section derived in this research was basically in compliance with the results reported in Xavier's study. No study has reported the variation in the shear moduli $G_{LT}$ and $G_{RT}$ along the whole cross-section of wood, whether dry or green. Therefore, the variation patterns of shear moduli $G_{LT}$ and $G_{RT}$ presented in this paper could be used to describe the shear properties of green wood in LT and RT plane. More data from identical or different species need to be obtained to determine the variation in shear properties of dry or green wood, especially for shear moduli $G_{LT}$ and $G_{RT}$.

### 3.3.3. Poisson's Ratio

The results from experimental data (Table 2) showed that the relationship between the values of the Poisson's ratios at the four sampling positions and sampling distance $R$ was relatively discrete, except for Poisson's ratio $\nu_{RT}$. To determine the quantitative relationship between Poisson's ratios and sampling distance $R$, three extra data points were inserted using interpolation for each Poisson's ratio, apart from $\nu_{RT}$. Thus, the relationships between the Poisson's ratios and sampling distance $R$ were obtained as shown in Figure 9a–f. The corresponding fitting equations are provided in Table 7.

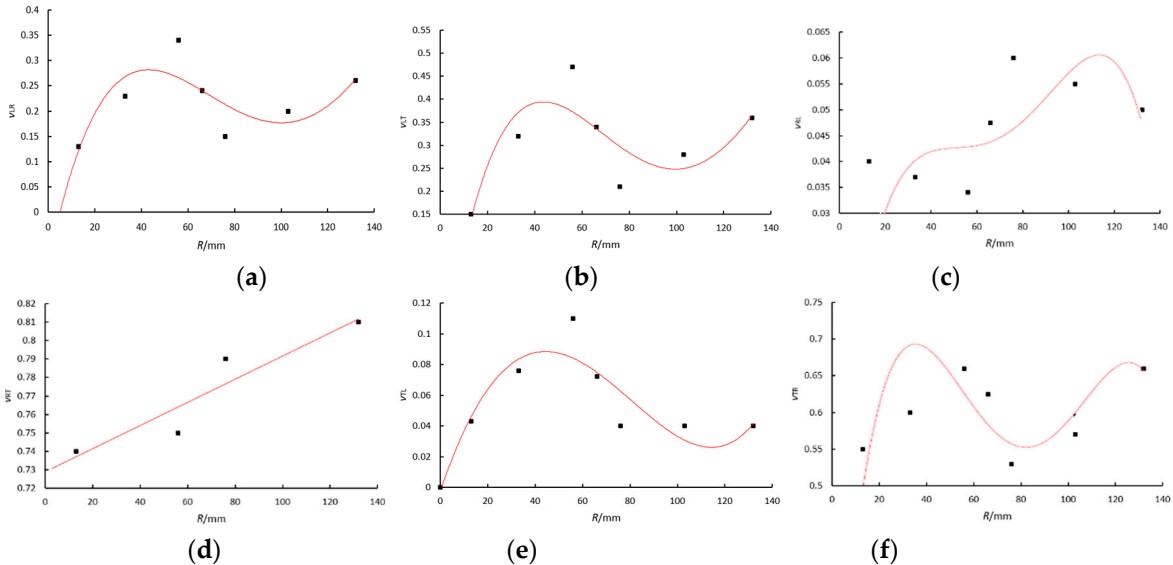

**Figure 9.** Relationships between distance from pith and Poisson's ratio of: (**a**) $v_{LR}$, (**b**) $v_{LT}$, (**c**) $v_{RL}$, (**d**) $v_{RT}$, (**e**) $v_{TL}$, and (**f**) $v_{TR}$.

Figure 9 shows the variation patterns of the six Poisson's ratios along the cross-section of the green wood. Figure 9d depicts the linear relationship between the Poisson's ratio $v_{RT}$ and sampling distance. Poisson's ratio $v_{RT}$ gradually increased with sampling distance. However, for the other five Poisson's ratios, there was a discrete relationship between the Poisson's ratio and sampling distance $R$. For Poisson's ratios $v_{LR}$, $v_{LT}$, and $v_{TR}$, the values first increased at sampling distances lower than 50 mm and then significantly decreased as sampling distance varied from 50 mm to about 80 mm. When the sampling distance was over 80 mm, the values of Poisson's ratio increased with sampling distance again. However, similar results were not found for Poisson's ratios $v_{RL}$ and $v_{TL}$. The values of the Poisson's ratios determined in this study irregularly changed with sampling distance probably due to the variation in moisture content, density, or microfibril angle in different parts of the wood. In general, no significant variation patterns were found in these five Poisson's ratios. Davies et al. estimated six Poisson's ratios both in outerwood and corewood from green *Pinus radiata* and no significant difference between outerwood and corewood was found [32]. Therefore, we still do not understand the variation patterns of the Poisson's ratios along wood cross-sections, and few researchers have evaluated the variation in the Poisson's ratios along the whole cross-section of dry or green wood. Therefore, more efforts are required to investigate the variation patterns of Poisson's ratio in the entire cross-section of dry or green wood.

**Table 7.** Mathematical model of Poisson's ratios and distance from pith.

| Poisson's Ratio | Fitted Equation |
|---|---|
| $v_{LR}$ | $v_{LR} = -0.0078 + 0.0138R - 2.112 \times 10^{-4}R^2 + 8.8643 \times 10^{-7}R^3 - 3.155 \times 10^{-8}R^4$ |
| $v_{LT}$ | $v_{LT} = -0.0124 + 0.0172R - 2.2671 \times 10^{-4}R^2 + 5.2757 \times 10^{-7}R^3 + 2.8007 \times 10^{-9}R^4$ |
| $v_{RL}$ | $v_{RL} = 0.0044 + 0.00276R - 7.1388 \times 10^{-5}R^2 + 7.7972 \times 10^{-7}R^3 + 2.8887 \times 10^{-9}R^4$ |
| $v_{RT}$ | $v_{RT} = 0.7312 + 6.2191R$ |
| $v_{TL}$ | $v_{TL} = -0.0023 + 0.0046R - 6.7872 \times 10^{-5}R^2 + 1.9557 \times 10^{-3}R^3 + 5.4162 \times 10^{-10}R^4$ |
| $v_{TR}$ | $v_{TR} = 0.028 + 0.0471R - 0.0011R^2 + 1.0274 \times 10^{-5}R^3 - 3.155 \times 10^{-8}R^4$ |

## 4. Conclusions

The objective of this study was to investigate the variation in the mechanical properties, especially elastic constants, of green Chinese larch from pith to sapwood. The conclusions are as follows:

(1) The relationships between longitudinal modulus of elasticity ($E_L$), radial modulus of elasticity ($E_R$), and tangential modulus of elasticity ($E_T$) were $E_L > E_R > E_T$ for all four sampling positions. Similarly, $v_{RT} > v_{LT} > v_{LR}$ were found for Poisson's ratios $v_{RT}$, $v_{LT}$, and $v_{LR}$ at the four sampling locations. These results align with the reported findings in dry wood.

(2) The sensitivity of each elastic constant to the sampling position was different, and the coefficient of variation ranged from 4.3% to 48.7%. The Poisson's ratios $v_{RT}$ measured at the four different sampling positions were similar and the differences between them were not significant. The coefficient of variation for Poisson's ratio $v_{RT}$ was only 4.3%. The four sampling positions had similar Poisson's ratios $v_{TL}$, though the coefficient of variation was 11.7%. The Poisson's ratio $v_{LT}$ had the greatest variation in all elastic constants with a 48.7% in coefficient of variation.

(3) We found a good linear relationship between the longitudinal modulus of elastic $E_L$, shear modulus of elasticity $G_{RT}$, Poisson's ratio $v_{RT}$ and the distance from pith. $E_L$, $G_{RT}$, and $v_{RT}$ all increased with sampling distance $R$. However, a quadratic relationship existed in the tangential modulus of elasticity $E_T$, radial modulus of elasticity $E_R$, shear modulus of elasticity $G_{LT}$, shear modulus of elasticity $G_{LR}$, and the distance from pith. A discrete relationship was found in the other five Poisson's ratios.

**Author Contributions:** F.L. wrote the original draft of the manuscript and performed most of test and analysis work. H.Z. and C.G. supervised the research team and provided some ideas to research as well as revised the manuscript. F.J. provided the main idea of experiments and assisted with the experiments. X.W. conceived the project and provided technical guidance to the research.

**Funding:** This project was supported by "the Fundamental Research Funds for the Central Universities (NO. BLX201817)", China Postdoctoral Science Foundation (NO. 2018M641225), the Special Research Funds for Public Welfare (NO. 201304512) and the National Natural Science Foundation of China (No. 31328005).

**Acknowledgments:** The authors wish to thank Jianzhong Zhang for his grateful assistance of conducting the experiments.

**Conflicts of Interest:** The authors declare no conflict of interest.

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
