# Peer review of "Variations in Orthotropic Elastic Constants of Green Chinese Larch from Pith to Sapwood"

_forests, doi:10.3390/f10050456_

Round 1
Reviewer 1 Report
Variations in material constants of green Chinese larch from pith to sapwood were investigated and analyzed in this paper. The paper is well written and structured. The novelty of the work is clearly presented. The topic is interesting. Minor corrections are requested to improve the paper:
1. Do not connect units with a number such as for example 310MPa, 53mm (check throughout the whole manuscript).
2. Knowledge about the mechanical properties of wood is very important. The authors used two Chinese larch trees aged 40 years. Why did they choose such age of the trees? Does it play any role in their research? Most of the strength properties of wood seem to change rather slowly with aging, or not at all.
3. The authors may also wish to give a more detailed discussion of Figures 9? Could do the authors explain the courses of declared graphs (for example Figure 9 b – it seems to be a sinuous course). Very little mention of explanation is made.
4. The authors write….“ More data from identical or different species need to be required to determined the variation in shear properties of dry or green wood, especially for shear moduli GLT and GRT.“ Are they going to make other research? Coefficient of determination R2 for GTR is quite low. Can you give better own interpretation of it?
Author Response
Response to Reviewer 1 Comments
Point 1: Variations in material constants of green Chinese larch from pith to sapwood were investigated and analyzed in this paper. The paper is well written and structured. The novelty of the work is clearly presented. The topic is interesting and deserves to be published. Only minor corrections are requested to improve the paper.
Response 1: Special thanks to you for the affirmation of our scientific research work and paper. We have carefully revised the manuscript based on your comments and recommendations.
Point 2: Do not connect units with a number such as for example 310MPa, 53mm (check throughout the whole manuscript).
Response 2: Corrected. The units and a number have been separated in the whole text.
Point 3: Knowledge about the mechanical properties of wood is very important. The authors used two Chinese larch trees aged 40 years. Why did they choose such age of the trees? Does it play any role in their research? Most of the strength properties of wood seem to change rather slowly with aging, or not at all.
Response 3: As mentioned in the article, Chinese larch is a common plantation species in northern China. Chinese larch with an age of 40 years is suitable to manufacture the wood products, such as furniture and musical instruments. Therefore, it is meaningful to determine the mechanical properties of 40 years Chinese larch trees in order to decide the final usage of larch wood. Additionally, the effect of tree age on mechanical performance of wood was not required to be discussed in this paper. The effect of age on the mechanical properties of Chinese larch trees may need to be further investigated.
Point 4: The authors may also wish to give a more detailed discussion of Figures 9? Could do the authors explain the courses of declared graphs (for example Figure 9 b – it seems to be a sinuous course). Very little mention of explanation is made.
Response 4: More detailed descriptions of Figure 9 have been added. It is really hard to provide the exact explanation or courses for Figure 9 because the mechanical properties of wood may be affected by many factors such as the density of wood, microfibril angle, internal defects, etc. The values of Poisson's ratios determined in this article changed with sampling distance probably due to the variation in density or microfibril angle in different part of wood.
Point 5: The authors write….“ More data from identical or different species need to be required to determine the variation in shear properties of dry or green wood, especially for shear moduli GLT and GRT.“ Are they going to make other research? Coefficient of determination R2 for GTR is quite low. Can you give better own interpretation of it?
Response 5: We will go to make further research to determine the variation in shear properties of green wood in different species, especially for shear moduli GLT and GRT. The possible interpretation for the relatively lower coefficient of determination R2 for GTR was on the one hand possibly due to insufficient sampling distance, and on the other hand probably attributed to the large variability of wood performance especially in cross section.

Reviewer 2 Report
This is manuscript is clear, sound, novel and interesting: compliments to the authors.
In the attached pdf are my comments: in light blue those referring to the content, in orange those referring to English language and writing. When reading it, please make sure you are seeing all comments: not all of them can be seen by just moving the cursor on the text highlighted (for instance, open the "Comments" menu to see all of them).
My comments regard minor modifications: improvement of English language; modification of Figure 2; considerations on how to comment on R2 values; other minor comments etc.
In my opinion, the manuscript can be accepted after the above minor revisions.

Author Response
Response to Reviewer 2 Comments
Point 1: This is manuscript is clear, sound, novel and interesting: compliments to the authors. In my opinion, the manuscript can be accepted after the above minor revisions.
Response 1: Special thanks to you for the affirmation of our scientific research work and paper.
Points 2: In the attached pdf are my comments: in light blue those referring to the content, in orange those referring to English language and writing. When reading it, please make sure you are seeing all comments: not all of them can be seen by just moving the cursor on the text highlighted (for instance, open the "Comments" menu to see all of them).
Response 2: We have carefully revised the manuscript based on your comments and recommendations.
Point 3: Delete ", such as wood,". I understand your point, but wood is well-known as anisotropic material. Phrases on its orthotropic behaviour on a specific direction should be carefully written to avoid confusion. (Section 1, Line 44)
Response 3: This is a good recommendation, ", such as wood," has been deleted.
Point 4: Proper terms are heartwood and sapwood. Did Davies et al. specifically refer to "corewood and outerwood" for some reason? If yes, please briefly explain in the text; if not, replace with "heartwood and sapwood". (Section 1, Line 89-90)
Response 4: Davies et al. specifically referred to corewood and outerwood in their paper. They would like to provide the material constants necessary for describing green corewood and outerwood as an orthotropic material, so that tree stems can be modelled more realistically.
Point 5: Replace with "Diameter at breast height (DBH)", so that the acronym is already introduced prior to row 136. (Section 2, Line 114)
Response 5: This is a very good recommendation, "Diameter at breast height" has been replaced with "Diameter at breast height (DBH)".
Point 6: Please briefly define what parameter R is. (Section 2, Line 133)
Response 6: Parameter R was referred to the distance between the centre of pith and the centre of sampling position, which has been defined in the text.
Point 7: Maybe "clean test specimens" is more appropriate, please verify and in case replace. (Section 2, Line 155)
Response 7: This is a very good recommendation, "defect-free test specimens" has been replaced with "clear test specimens".
Point 8: Replace with "is". Reference to figure and tables should be in present tense. Please check the entire text for such replacement (see for instance rows 223, 282, 325, 333). (Section 2, Line 200)
Response 8: This is a very good recommendation. The tense referred to figure and tables has been revised and used in present tense throughout the entire text.
Point 9: See the precedent comment about "corewood" and "outerwood". (Section 3, Line 226)
Response 9: The terms used in Davies’ paper were exactly corewood and outerwood.
Point 10: Results of GRT, GLR and GLT are not discussed here in the text but later in 3.3.2. Maybe something can be added here, referring to 3.3.2, since the values are shown above in Table 2. (Section 3, Line 260)
Response 10: The discussion about the results of shear modulus GRT, GLR and GLT has been added in the text.
Point 11: Replace with "As already mentioned, in many studies wood is considered as an orthotropic material in the three main orthotropic directions". See comments at rows 38-44. (Section 3, Line 266)
Response 11: This is a good recommendation, and this sentence has been replaced.
Point 12: A short table with the values of Eq. (6) and (7) for P1, P2, P3 and P4 could be added. (Section 3, Line 271-272)
Response 12: This is a very good recommendation. Elastic constants obtained from P1, P2, P3 and P4 were proved that they work really well for Equation (6) and (7). Taking Equation (7) for an example, a short table with the values of Eq. (7) for P1, P2, P3 and P4 has been added in the text.
Point 13: Why could? I would delete could, leaving "from P1 satisfy". (Section 3, Line 272)
Response 13: This is a good recommendation. “Could” has been deleted in the text.
Point 14: Comments on correlations (that is on R^2 values) shall be made in comparison to other reference values from literature. Phrases such as "Strong correlation" (row 300) or "good linear relationship" (row 334) shall be completed with a term of comparison. For instance, a R^2 value of 0.90, which seems high, could be actually low in a physic experiment performed in a strictly controlled environment where usually R^2 values obtained are higher than 0.95.To mention some general examples, comments on correlation shall be "a R^2 value of ..." or "a high value compared to this study..." or "a value in line with that found by...". Please check the entire text for this type of modification. (Section 3, Line 300)
Response 14: This is a very good recommendation. Comments on correlations, i.e. R2 value, have been modified in the text.
Point 15: Modification of Figure 2.
Response 15: Figure 2 has been modified.
Other changes:
We have done some corrections in wording and sentence structure in the other part of the manuscript. Revised portion are highlighted in yellow in the manuscript.

Reviewer 3 Report
This article is not in the scope of the journal, it is more technical, without any background informations about source of timber - e.g. climate, and so on.
Author Response
Response to Reviewer 3 Comments
Point 1: This article is not in the scope of the journal, it is more technical, without any background information about source of timber - e.g. climate, and so on.
Response 1: Special thanks to your comments. Firstly, this article was to investigate the mechanical properties of green wood and therefore it should be in the scope of special issue “Wood Properties and Processing”, which belongs to the section “Forest Ecophysiology and Biology” in Forests. The interests of this article were in line with that of special issue “Wood Properties and Processing”, i.e. wood performance. Moreover, the keywords of special issue “Wood Properties and Processing”, such as wood, performance and mechanical properties, were consistent with the keywords of this article. Secondly, the background information about source of timber has been provided in Materials section (Section 2, Line 110-112). We have provided the longitude and latitude of plantation, thereby, the climate of plantation could be obtained subsequently.

Reviewer 4 Report
Dear authors,
you have done a very good job. The paper seems to be professional and we would like to congratulate you.
Suggestions:
LIne 32-33 Please revise this general sentence enrich it with more detailed characteristics or delete it.
LIne 411 Please change this very trivial explanation of the results to more scientific.
Author Response
Response to Reviewer 4 Comments
Point 1: you have done a very good job. The paper seems to be professional and we would like to congratulate you.
Response 1: Special thanks to you for the affirmation of our scientific research work and paper.
Points 2: Line 32-33 Please revise this general sentence enrich it with more detailed characteristics or delete it. (Section 1, Line 32-33)
Response 2: This sentence has been revised and enriched with more detailed characteristics.
Point 3: Line 411 Please change this very trivial explanation of the results to more scientific. (Section 4, Line 411)
Response 3: The first part of conclusion has been changed to more scientific.
